# A Systematic Study on Impact of Binder Formulation on Green Body Strength of Vat-Photopolymerisation 3D Printed Silica Ceramics Used in Investment Casting

**DOI:** 10.3390/polym15143141

**Published:** 2023-07-24

**Authors:** Ozkan Basar, Varghese Paul Veliyath, Fatih Tarak, Ehsan Sabet

**Affiliations:** 1Wolfson School of Mechanical, Electrical & Manufacturing Engineering, Loughborough University, Loughborough LE11 3TU, UK; b.ozkan@lboro.ac.uk (O.B.);; 2Additive Manufacturing Centre of Excellence, 33 Shaftesbury Street South, Derby DE23 8YH, UK

**Keywords:** vat photopolymerisation, ceramics, ceramic core, investment casting, 3D printing ceramics

## Abstract

Additive ceramics manufacturing with vat-photopolymerisation (VP) is a developing field, and the need for suitable printing materials hinders its fast growth. Binder mixtures significantly influence the mechanical properties of printed ceramic bodies by VP, considering their rheological properties, curing performances and green body characteristics. Improving mechanical characteristics and reducing cracks during printing and post-processes is mainly related to binder formulations. The study aims to develop a binder formulation to provide the printed ceramic specimens with additional green strength. The impact on mechanical properties (ultimate tensile strength, flexural strength, Young’s and strain at breakpoint), viscosity and cure performance of Urethane Acrylate (UA) and thermoplastic Polyether Acrylate (PEA) oligomers to monofunctional N-Vinylpyrrolidone (NVP), 1,6-Hexanediol Diacrylate (HDDA) and Tri-functional Photocentric 34 (PC34) monomers were investigated under varying concentrations. The best mechanical characteristic was showcased when the PC34 was replaced with 20–30 wt.% of UA in the organic medium. The Thermogravimetric Analysis (TGA) and sintering test outcomes revealed that increasing the content of NVP in the organic medium (above 15 wt.%) leads to uncontrolled thermal degradation during debinding and defects on ceramic parts after sintering. The negative effect of UA on the viscosity of ceramic-loaded mixtures was controlled by eliminating the PC34 compound with NVP and HDDA, and optimum mechanical properties were achieved at 15 wt.% of NVP and 65 wt.% of HDDA. PEA is added to provide additional flexibility to the ceramic parts. It was found that strain and other mechanical parameters peaked at 15 wt.% of PEA. The study formulated the most suitable binder formulation on the green body strength of printing silica ceramics as 50 wt.% HDDA, 20 wt.% Urethane Acrylate, 15 wt.% NVP and 15 wt.% PEA.

## 1. Introduction

Vat photopolymerisation (VP) is the most prominent and accurate process compared to other additive manufacturing (AM) techniques since the process can fabricate complex and intricate parts with exact tolerances and high surface quality at excellent resolutions down to the micrometre range [1,2]. Digital light processing (DLP) and liquid crystal display (LCD) are the two recently developed mask-based VP technologies, after stereolithography (SLA). In both processes, the light source beams a single image of each layer by making the pixels appear bright/dark on the projection screen, forming the whole layer simultaneously instead of drawing rounded lines as SLA does [3]. LCD technology has recently gained significant popularity in ceramic printing applications and offers a fast and cost-efficient alternative to DLP and SLA systems [4]. The light emitted by the LCD screen initiates a photochemical reaction (free radical polymerisation) in a vat containing mainly a ceramic-loaded (meth)acrylate or epoxy photocurable monomer resin in tandem with other additives such as photo-/co-initiator system.

The number of reactive functional groups and chemical structure of monomers have a decisive impact on their viscosity, reactivity and cross-linking density in the respective ceramic-loaded system [5]. Monofunctional monomers (named reactive diluents due to their low viscosity) are formed by linear chains in which the parts made with entangled linear monomers do not exhibit good mechanical properties for both polymers and ceramics [6,7]. In contrast, multifunctional monomers can generate covalent bonds among molecules, resulting in cross-linked networks in non-linear shaped forms during polymerisation [8], enhancing the printed part’s overall part strength and hardness. Achieving higher solid loading and green body strength in all photoinduced ceramic processes is fundamental to obtaining optimum densification and mechanical properties on sintered bodies [9]. However, the greater the viscosity of the monomer formulation (i.e., the molecular weight of monomers thus increases chain length or functionality) the lower the dispersibility of ceramic particles [10]. Since the viscosity of ceramic suspensions is proportional to the binder viscosity multiplied by the impact of solid loading, the medium viscosity should be as low as possible to achieve high solid loading with moderate viscosity in the final mix. Thus, highlighting the significance of the binder formulation used in ceramic suspensions is the critical factor determining the success and quality of the prints.

Generally, 1,6-Hexanediol diacrylate (HDDA) is one of the most generic monomers used in ceramic VP applications due to its low viscosity and good cross-linking behaviour [11,12]. The mechanical properties of ceramic green bodies can be further improved by mixing HDDA with (meth)acrylate monomers from various functional groups [13,14,15,16,17,18], and the overall viscosity of the medium can be reduced by reactive diluents such as N-Vinylpyrrolidone (NVP) [4], or acryloyl morpholine (ACMO) [19]. Although print quality and accuracy significantly depend on the viscosity degree of mixtures, the green body strength is another vital factor due to the high peeling forces, such as printing parts in the vertical direction with a viscosity of up to 10,000 mPa·s [20]. The parts made by most AM technologies are strong in X- and Y- axes and weak in the Z-axis due to the weak bond formation between layers [21,22]. As the Z-axis would be considered the height of the printed part, tensile strength is weaker along the Z-axis, especially in more elevated printed ceramic parts. Low green strength can affect the part’s ability to hold shape during printing and generate cracks under high peeling forces [23]. Low strength can also split interlayer bonding regions and crack under thermal stress while sintering [24,25]. It is, therefore, essential to investigate the binder formulations and factors affecting the green strength to reduce the impact of peeling forces seen in ceramic printing applications.

Proper selection of the oligomeric backbone structure in the organic medium is vital in formulating rigid and flexible 3D printing resin/inks with superior mechanical properties. Each kind of oligomer with a different chain structure has a different advantage. For example, aliphatic urethane acrylate oligomers are ideal for excellent cross-linking during photopolymerisation due to the hydrogen bonds formed by N-H and C=O, thus providing increased tensile strength and strain to the resins [26,27]. In contrast, the continuous modification of the polymer chain structure makes the polyester/polyether oligomers more and more functional for improving the flexibility and adhesion between cured layers.

Despite a handful of studies on the optimisation and characterisation of monomers as ceramic binders [4,19,28,29], a recent review by Leite de Camargo et al. [12] shows that the effect of oligomers on the viscosity of the ceramic binder systems, curing performance (cure depth) and mechanical properties of printed bodies have not been fully studied so far. Therefore, we aim to extend the investigation conducted in the previous study [4] by understanding the effect of varying the concentrations of the aliphatic urethane and polyester acrylates on the viscosity of the mix, curing performance of the formulation in the LCD printing process, as well as on the mechanical properties of ceramic prints. The study systematically explores the relationship between oligomer and mono and multifunctional monomer mixtures, their cross-link density, rheological behaviour, mechanical properties and their performance as binders in ceramic vat photopolymerisation. This study aims to develop a binder formulation for ceramics that provides substantial mechanical properties at high solid content with the lowest possible viscosity while avoiding the build-up of internal stresses and crack for the ceramic parts that need to be printed vertically, such as ceramic cores.

## 2. Materials and Methods

### 2.1. Materials

A set of (meth)acrylate monomers and oligomers with different degrees of functionality are used to formulate photopolymer binders in this study. HDDA and NVP were utilised in all mixes as reactive diluents and a trifunctional Photocentric 34 (PC34) monomer was used to achieve good offset properties and excellent reactivity [4]. A difunctional aliphatic urethane acrylate (UA), diluted with 25% of Dipropylene Glycol Diacrylate (DPGDA), is used as an oligomer to exhibit high flexibility, toughness and adhesion for the binder formulations. A reactive trifunctional polyether acrylate (PEA) is employed to enhance the flexibility of the binder formulations. Rahn Ltd. (London, UK) and AM Centre of Excellence Ltd. (Derby, UK) provided all chemical compounds of the binder formulations used in this study. The summary of material properties of all monomer/oligomer compounds with their properties and labelled names is demonstrated in Table 1. A silica-based ceramic powder mix (with a d_50_ particle size of 5 µm) and an active polymeric dispersant were infused with all formulated binder mixtures to test their impact on the rheological behaviour of suspensions and mechanical properties of printed parts. A complete study of the properties and sources of the powder mixture and commercial dispersant is published elsewhere [20].

### 2.2. Materials Preparation

2,4,6-trimethyl benzoyl phosphine oxide (BAPO) was used to achieve a typical free radical photopolymerisation of ceramic suspensions, mixed with a 1.0 wt.% of all binder mixtures, stirred using a magnetic stirrer for 25 min. The complete dissolution of the photoinitiator in all mixes was ensured by sonicating in an ultrasonic bath at 60 °C for 1 h. The ceramic powder mixture was dehumidified in an oven at 80 ºC for 24 h prior to loading into the binder mixtures. The dispersant agent was dissolved in pre-prepared photoreactive binder mixtures in 0.5 wt.% of the ceramic powder and the desired powder content was then added to the medium. A dual asymmetric centrifugal mixer (SpeedMixer DAC 400 FVZ, Hauschild, Hamn, Germany) was used to ensure sufficient dispersion at 2000 rpm for 15 min, followed by 10 min at 400 rpm to complete dispersion without agglomerates.

### 2.3. Characterisation of Mixtures before and after 3D Printing

The Cure depth (C_d_) of the prepared binder mixtures was measured by exposing tiny droplets of the samples to ultraviolet (UV) light for a fixed time using an Anycubic Photon S LCD-based 3D printer (Anycubic Technology Co., Ltd., Hongkong, China) with a 405 nm wavelength and light intensity 0.44 mW/cm^2^. The droplets of samples were extracted using a pipette and were poured onto a 100 µm thick Fluorinated Ethylene Propylene (FEP) film and exposed to UV light from the printer. All the samples were investigated for 8 s, 10 s, 12 s, 15 s, 17 s and 20 s of exposure time to identify their cure depth. An external 0–25 mm micrometre (Mitutoyo Corp, Kawasaki, Japan) was used to measure four individual thickness measurements for all samples for each exposure time. The average of these measured thicknesses was taken as the penetration depth (Dp) for each light exposure energy (E_c_). The above values were used in Jacob’s basic working equation [30] to find the optimum printing time; detailed information on C_d_ measurements and identification of printing parameters are provided in the Appendix A (Appendix A).

The viscosity of the binder mixtures was measured using Ametek Brookfield RST Rheometer (AMETEK GB Ltd. T/A Brookfield, Harlow, UK). The test is carried out by adding a small amount of the prepared binder mixture onto a temperature-controlled static stainless steel holding plate which is then exposed to a constant shear rate of 800 s^−1^ at a temperature of 25 °C by a rotating 50 mm parallel stainless-steel plate at a gap of 0.2 mm. The average of the viscosity readings from the test is taken as the viscosity of the binder mixture. For the viscosity measurements of ceramic-loaded mixtures, less than 2.0 mL of the sample was loaded onto the static holding plate with a gap of 0.5 mm to the opposite stainless-steel parallel plate. To allow the rheometer to recalibrate and reduce the effects of previous shear measurements, the samples were left for 2.5 min before the next set of viscosity measurements. The viscosity of suspensions as a function of shear rate from 0 to 200 s^−1^ was measured at 25 °C.

All test samples for all un- and loaded mixtures were printed in the Z (vertical) direction by 3D printer at room temperature. The mechanical properties of all binder mixtures were studied, according to ASTM standard D638 [31], using a Mecmesin Multitest-dV Force tester (PPT Group Ltd., Slinfold, UK) with a capacity of 2.5 kN at the ASTM standard speed of 0.5 mm/min. The ASTM standard test method C1161-18 was used to measure the flexural strength of all green (not-sintered) ceramic rectangular test bars with a dimension of 4 mm × 3 mm × 45 mm (type B). The distance between two supports in the three-point bending test was 40 mm, the loading span was 10 mm and the crosshead speed of the flexural strength test was controlled at 0.5 mm/min. Both tensile and flexural test samples were tested right after printing to minimise the deviations in mechanical properties due to post-curing. Ten measurements were recorded, averaged and then plotted on each set.

The Thermogravimetric Analysis (TGA) is used to define the thermal decomposition characteristics of binder formulations, which are crucial to define de-binding and sintering post processes. TA Instruments Q5000IR (TA instruments, New Castle, DE, USA) testing device was used between temperature ranges of 30 °C up to 600 °C with a ramp-up temperature of 0.5 °C/min to determine the characteristic of selected combinations in ambient air.

## 3. Results and Discussion

### 3.1. Improving the Mechanical Performance of Binder System for Ceramics 

The previous literature [4] was focused on identifying the impact of the di- and multifunctional binder materials on the printed parts for which the best performance (mechanical properties and rheological behaviour) has been achieved when the difunctional (HDDA) and multifunctional (PC34) monomers were mixed in a 50:50 ratio. The addition of monofunctional monomers (NVP) in the organic medium proved to be a significant improvement in the rheological behaviour of loaded suspensions and mechanical properties of printed samples [9]. Hence, the formulation consisting of 5 wt.% NVP, 45 wt.% HDDA and 50 wt.% PC34 has been selected to further investigate in the present work and divided into four stages. 

The first part of the research (Section 3.1.1) aims to understand the impact of different weight percentages of UA as an oligomer by replacing the weight of the PC34 compound in the selected mixture. The second study (Section 3.1.2) explores the impact of varying the proportion of the reactive dilutants NVP and HDDA on the mixtures formulated with UA. After identifying optimum UA and NVP contents based on the outcomes of the first two studies, the third part (Section 3.1.3) aims to identify the influence of eliminating the PC34 compound with HDDA on the mechanical and rheological properties. The results of cure depths, viscosities and mechanical properties of individual binder mix, including their impact on ceramic-loaded mixtures, are discussed in the sections below.

#### 3.1.1. Impact of Aliphatic Urethane Acrylate

The impact of UA on the mechanical and rheological properties was investigated by replacing the PC34 compound with UA between 5 and40 wt.%. The cured thickness under fixed exposure energy increases with the increase of oligomer concentration (Figure 1). The increase in cure thickness of blends could be attributed to the presence of hydrogen bonds in the hard points of the UA and good agreement in previous studies [32]. The C_d_ of the existing mixture from the previous study [4] was improved by incorporating UA and reducing the PC34 monomer from the trifunctional group.

Figure 2a demonstrates the mechanical performance outcomes of the binder from the tensile test. The UTS increases gradually with UA addition and peaks at 20 and 30 wt.%, respectively. Although there is only a negligible difference in monomer blends’ UTS at 20−30 wt.% concentration, an increase of almost 125% in tensile strength is observed when the selected mixture [4] is modified by adding 20−30 wt.% of oligomers. In both tests, the strain increases gradually up to 40 wt.% of UA. Improving the oligomer content decreases Young’s modulus of the binder system significantly and makes the printed parts more flexible.

The flexural test of the ceramic samples illustrated a similar observation as made for the tensile test (Figure 2b). As expected, the flexural strength and strain increased gradually with the addition of UA and reached the maximum at 20−30 wt.%. With the addition of 40 wt.% oligomer, the drop in flexural strength by 21% is due to the high density of polymer cross-linking, which leads to the formation of a brittle structure in the cured parts, which is not desirable for vertical printing direction. Moreover, the flexural strain observed at the breaking point increased exponentially with the addition of oligomer due to the increasing presence of strong intermolecular forces (here, Hydrogen bonds), which helps in the stretchability observed in the above binder mixtures. The energy released during the breakage of hydrogen bonds when the medium is subjected to mechanical loading can be the reason behind the increasing strain [17,20].

The viscosity measurements in Figure 2a prove how the oligomers impact the binder’s viscosity, thus leading to a substantial increase in the viscosity of the ceramic-loaded medium. Figure 3a shows the viscosity of the 75 wt.% loaded ceramic ink with increased oligomer content in the mixture at a specific shear rate (here, 100 s^−1^). Figure 3b shows the difference in viscosity at different shear rates between the mix with the maximum and minimum oligomer content. The trend in viscosity is observed due to UA’s high molecular weight and intermolecular forces (due to the presence of hydrogen bonds). It has also been observed that the shear thinning behaviour (decrease in viscosity with the increase in shear rate) becomes more evident with the rise in oligomer content in the mixture (Figure 3b).

Considering the above set of test results, binder mixtures with 20 wt.% and 30 wt.% oligomers are selected to study further. The cure depth study proves the effect of polymerisation on mechanical properties with the addition of UA. However, high polymerisation can inhibit the purpose of adding oligomers by making the structure of the cured binder brittle. Figure 4 depicts an example of a vertically 3D-printed tall ceramic core model with the B-37 and 30 wt.% of UA-formulated mixtures; the part with B-37 loaded mixture failed after reaching a certain height due to low green strength and flexibility, whereas the same component printed successfully with the oligomer-formulated formulation.

High-functionality monomers’ influence on the mechanical properties of green bodies was studied in previous studies [19,29] with various methods. Even with different approaches such as individual investigation or mixture design strategy to understand the mechanical properties and shrinkage of green parts with high functional groups, similar results were demonstrated on the mechanical and rheological properties. As illustrated in Figure 1, due to the high polymerisation rate, UA−based ceramic formulations are favourable in different applications, such as electronics [33]. By prioritising mechanical properties to select optimum oligomer concentration, 20 and 30 wt.% were selected based on their similar mechanical properties and viscosity compared to other binder mixtures. Since their viscosity is significantly higher than the base formulation, the following two sections focus on dropping the viscosity and enhancing the mobility of reactive species in organic mediums by varying the NVP and HDDA content.

#### 3.1.2. Effect of NVP Concentration on Selected Mixes 

NVP is one of the reactive diluents among monofunctional UV-curable monomers in the market that helps to reduce the binder’s overall viscosity and improves the polymerisation rate [4,12]; therefore, the NVP is replaced incrementally by HDDA in this study. The study showed that increasing the content of NVP did not significantly impact the polymerisation rate of both organic mediums. The curing performance of the binder system containing 20 wt.% of UA improved when the NVP amount increased to 25 wt.%, with better polymerisation reaction at a higher exposure energy region due to increased mobility of reactive species (Figure 5a). For the binder system consisting of 30 wt.% of UA, the formulated mixtures with 10 and 25 wt.% of NVP showed a similar trend to 5 wt.% and the best performance at a high exposure energy, whereas the medium containing 40 wt.% of NVP had better curing performance at a low exposure energy (Figure 5b), and good agreement with the previous study [4].

Increasing the NVP content in the 20 wt.% of UA did not significantly impact the binder’s mechanical and viscosity properties, showing a fluctuating pattern with a peak of 18.36 MPa by 25 wt.% of NVP content (Figure 6a). With the addition of NVP into the 30 wt.% of UA counterpart, the tensile strength increased with a peak at 15 wt.% of NVP, followed by a steady decline from 15 to 40 wt.% of NVP (Figure 6b). It is evident that reducing the viscosity of the binder mixture with high oligomer content by NVP in a range of 10−25 wt.% increases the polymer chains’ mobility, thus improving the mechanical properties of printed parts. For both selected systems, the drop in tensile strength at 40 wt.% of NVP can be attributed to lowering the overall functionality of the respective binder formulation. A similar trend in Young’s Modulus and tensile strain is seen for both binder mixtures with increased NVP content; the cured binder becomes stiffer from 5 to 15 wt.% of NVP and becomes more flexible between 15 and 40 wt.% of NVP due to the decreasing Young’s modulus.

The tensile strain showed a similar trend for both UA content mixtures. Even the increase of tensile strain is minor until 15 wt.% of NVP concentration; tensile strain rises dramatically with more NVP amount. It is essential to note that tensile strain values are almost doubled with the 30 wt.% of UA content (Figure 6). Ceramic green bodies’ mechanical properties show results similar to the tensile outcomes, as shown in Figure 7. The flexural strain of both systems rose significantly after 15 wt.% of NVP, suggesting the flexibility between cured layers and adhesion to the build platform for the ceramic parts in vertical printing direction can be significantly improved by NVP in UA formulated binder mixtures.

Figure 8 demonstrates the decrease in the viscosity of ceramic suspensions with a rising amount of NVP. Both loaded mediums showed the same incremental reduction in the viscosity of mixes; their viscosity dropped about 1.5 Pa·s at 40 wt.% of NVP. This experimental outcome shows that increasing the content of reactive diluents in the mixtures with high UA content did not significantly impact the dispersibility of ceramics.

A recent study [19] demonstrated a similar pattern between monofunctional and multifunctional binder components, increasing the content of monofunctional monomer in the higher functional medium until a specific ratio improves the rheological behaviour of suspensions and mechanical properties of parts. Overall, NVP demonstrated a more prominent effect on the mechanical properties of ceramic green bodies than the suspensions’ viscosity in this study. Both selected mixtures’ functionality and cross-linking density dropped significantly when the NVP amount exceeded 25 wt.%.

It should be noted that having high control over the degradation of binders is vital during the high-temperature sintering process. Therefore, the degradation chemistry of reactants and the mass transport process are critical factors while designing the binder formulation, which requires a delicate balance between the volatilisation (pyrolysis rate) of organic compounds and their removal rates to avoid the formation of defects or cracks. Although the addition of oligomer suppresses the degradation rate of monomers (hence, early pore development), relieving high-pressure build-up within the porous body, it is difficult to control the thermal degradation of NVP compound with various additions discussed for different applications [20,23]. To this extend, the binder mixtures with varying concentrations of NVP were subjected to TGA to understand the impact of NVP on the thermal stability of ceramic parts. 

Figure 9a shows the exothermic peaks of binder mixtures with the derivative of weight (DW) curves confirming where mass reduction occurs in the sample. The sharpness of the peak shows that a significant fraction of the NVP compound in the binder mixtures is burned away at a specific temperature range (~420 °C). The sharp rise and high mass loss during the degradation were also noticed in related studies [34,35]. Therefore, to better understand the impact of the high mass loss of the NVP compound on ceramic bodies at a narrow temperature range (420 °C), the ceramic parts were debindered and sintered at 1250 °C for 6 h. The sintering regime of test bars can be found in the previous study [20]. The debinding hold time of each combination was set to 4 h at 150, 330 and 420 °C to decompose all organic materials sufficiently with a 0.5 °C/min heating rate to each target decomposition temperature. In both UA binder formulations, the sintered parts did not show any sign of cracking when the content of NVP did not exceed 15 wt.% (Figure 9b,c). However, as shown in Figure 9d,e, the sudden loss of the NVP compound puts significant pressure on the ceramic printed parts prepared with 25 and 40 wt.% of NVP, leading to the generation of cracks during the debinding process. It is a fact that, specifically when NVP content is lower than 15 wt. %, the smoothness of the peak in Figure 9a shows that burning over a range of temperatures can help reduce the generation of cracks during post-processes.

Although adding 25 wt.% of NVP in both selected binders improves the mechanical and rheological properties, the ceramic parts printed at a higher (>25 wt.%) NVP concentration demonstrate crack propagation after sintering. Overall, 15 wt.% of NVP amount was selected due to adequate mechanical, rheological and thermal performance, and 5 wt.% of NVP content was used to understand the effect of NVP for further material investigations in the next section.

#### 3.1.3. Impact of HDDA Concentration on Selected Mixes

In this section, the UA content is fixed to 20 and 30 wt.%, and the tri-functional monomer (PC34) is completely replaced by HDDA under varying NVP concentrations to identify the full impact of HDDA as a reactive diluent. The formulated binder mixes with their mechanical and viscosity properties for this study are summarised in Table 2.

The impact of HDDA on curing performance depends on the UA concentration; even the reactive index of HDDA is sufficient for vat-photopolymerisation. For binders with 20 wt.% of UA, when the HDDA content increases to 75 wt.% in the mixture, the cure depth of the binder mixtures reduces (Figure 10a). A similar trend could be seen in the 30 wt.% formulations (Figure 10b). This reduction in C_d_ is due to replacing a tri-functional monomer with a difunctional monomer which diminishes the overall polymerisation and cross-linking and thereby reduces the cured thickness under fixed exposure energy.

For binders with 20 wt.% and 30 wt.% UA, tensile strain reduces with the increasing content of HDDA, and tensile strength reaches the peak point by 65 wt.% and 55 wt.% of HDDA, respectively (Table 2). The Young’s modulus performs a similar pattern as UTS in both UA formulations. It is evident that HDDA has a significant impact on the flexural strain of ceramic suspensions, in which ceramic green parts became more brittle with the increase of HDDA content in both mediums, as shown in Figure 11a,b. The flexural test bars also showcase a similar trend in the properties of binders with both UA contents; when the HDDA content is increased to the average ratio, the flexural strength increases and then drops with further addition of HDDA (Figure 11a,b). High cross−linking density and mobility of reactive species explain the high performance of the cured parts in tensile and flexural tests, as mentioned in previous sections. On the other hand, the poor performance of cured binders at a higher HDDA content is due to high shrinkage and internal stresses [20]. It is to be noted that the second−best mechanical test result is showcased by 40 wt.% and 65 wt.% of HDDA for the mixes with 20 and 30 wt.% of UA, respectively. This experimental outcome shows the importance of achieving the optimum balance between high cross−linking density and the mobility of reactive species via oligomers and diluents, respectively.

Replacing PC34 with HDDA significantly drops the viscosity of 75 wt.% of silica-loaded suspensions. Even the change of the viscosity for 20 wt.% of UA binders was negligible (Figure 11a), the viscosity increased significantly (to 8.5 Pa·s) with the addition of HDDA from 30 wt.% to 55 wt.%, further addition in HDDA content decreased the suspension viscosity again (Figure 11b). This outcome shows that NVP has a more prominent effect than HDDA on the dilution of UA, thus dispersibility of ceramic−loaded systems.

Fundamental drawbacks of HDDA are low cure depth and high shrinkage, which need to be controlled with other monomers from different functional groups [12]. Binder material investigation in the literature [4,19] with different multi and difunctional monomers proves the trade-off relation between mechanical properties and rheological characteristics—the study conducted by H. Chen et al. [29] showed that high tensile strength and low viscosity reached between 48 wt.% and 70 wt.% of HDDA formulation, which has good agreement with this research. Considering the remarkable viscosity reduction in ceramic loading and better mechanical properties, the binder formulation consisting of 65 wt.% of HDDA, 20 wt.% of UA and 15 wt.% of NVP is selected as the most promising candidate among all formulations. However, as its strain rate was reduced because of the removal of the tri-functional monomer (PC34), the final study aimed to increase the strain rate of printed parts through polyether acrylate (PEA) and identify the optimum content in the mixture.

### 3.2. Impact of Polyether Acrylate (PEA) on Printing

The final study aims to investigate the impact of PEA addition on the rheological and mechanical properties of selected binder mixtures from the previous study (Section 3.1.3). The implications of PEA in the binder system are evaluated by replacing the HDDA with PEA at various concentrations (from 5 wt.% to 20 wt.%). The exposure study showed different patterns depending on the concentration of PEA, which does not considerably affect the curing depth, as shown in Figure 12. The binder system with 15 wt.% of PEA showed the best cure performance at higher exposure energy, followed by 10 wt.%, 5 wt.% and 20 wt.%.

Figure 13a illustrates the impact of increasing PEA on unloaded mixtures’ mechanical and viscosity behaviour. As expected, Young’s modulus decreases and tensile strain increases gradually with the total amount of PEA in the mixture. Increasing the content of PEA did not significantly impact the tensile strength of parts, and the highest value was obtained with 15 wt.% of PEA. For the silica-loaded suspensions, as PEA’s concentration increases from 5 wt.% to 20 wt.% in the mixture, the flexural strength reduces slightly to 15 MPa, whereas flexural strain gradually increases (Figure 13b). PEA is a highly viscous material (as seen in Table 1) and, therefore, can significantly impact the viscosity of the binder system. As the PEA content is increased from 5 wt.% to 20 wt.%, there is a gradual increase in the viscosity of the binder mixture (Figure 13a). A similar rheological pattern is monitored on ceramic-infused binder mixtures, as seen in Figure 13b. 

However, replacing the HDDA content with PEA up to 15 wt.% did not substantially impact the dispersibility of the ceramic mixtures as the viscosity increase in the loaded medium was only about 2 Pa·s at 15 wt.% of PEA. The remarkable impact of PEA on the viscosity of suspension was seen when the content of PEA exceeded 15 wt.% with a 50% increase in the viscosity of the ceramic suspension.

The research monitored a similar trend to the previous study conducted by Badev et al. [36], in which the substitution of HDDA by PEA declines the overall functionality of the system and the network density. This results in a lower excess of free volume, which does not favour increasing the flexural strength of ceramic parts. Although adding 15 wt.% of PEA reduces the flexural strength of ceramic parts and increases the viscosity of the suspension, its impact on both properties was negligible and acceptable for printing silica parts. Considering the favourable effect of PEA on the flexibility (strain rate) of printed parts, the best-performing binder with 15 wt.% of PEA is selected in the final part of this study.

The printability of large and complex core models used in industrial gas turbines was investigated to validate the results of this study and the feasibility and scalability of the developed binder system for ceramic core printing. Figure 14 demonstrates the successful printing and sintering outcomes of these large complex−shaped ceramic core models used in industrial gas turbine applications without any crack, highlighting the significance of the binder’s mechanical properties and rheological behaviour for the bottom−up vertical ceramic printing approach.

## 4. Conclusions

In this work, each binder component is optimised to develop a formulation showcasing better mechanical properties and green strength than the reference mixture from previous research [4]. The optimisation process was conducted with four steps focused on each binder ingredient. Incorporating the UA into the formulated binder system significantly impacts the binder’s curing depth, viscosity and mechanical properties. Even the curing depth improves with increasing the concentration of UA; its high network density leads to the formation of a brittle structure when UA is used excessively in the binder system. The test outcome shows that the optimum amount is between 20 and 30 wt.% of UA due to the best mechanical properties achieved. In order to control the negative impact of UA on the viscosity of ceramic suspensions, the NVP and HDDA content in the binder mixtures increased by eliminating PC34 trifunctional compound. The results demonstrated a remarkable viscosity reduction in the ceramic-loaded mediums, thus leading to better mechanical properties on ceramic green bodies due to the increased mobility during polymerisation. Although increasing the concentration of NVP up to 25 wt.% leads to a substantial improvement in the rheological and mechanical properties of binder mixtures, its sudden loss during debinding puts considerable pressure on the ceramic parts printed at higher (>25 wt.%) NVP concentration, leading to crack generation after sintering. The research suggests the optimum amount of NVP and HDDA as 15 and 55–65 wt.%, respectively. 

On the other hand, multifunctional monomer PEA was added to the developed mixture to increase the flexibility of ceramic bodies. The reason behind the addition is to reach the best properties in terms of viscosity, tensile strength and flexural strain. Thus, considering the reaching high curing performance, desired rheological properties and mechanical properties, the best-performing binder formulation is 50 wt.% HDDA, 530 20 wt.% Urethane Acrylate, 15 wt.% NVP and 15 wt.% PEA. Ceramic cores with different dimensions, designs and infills are printed successfully as proof of reached performance of the developed ceramic suspension.

## Figures and Tables

**Figure 1 polymers-15-03141-f001:**
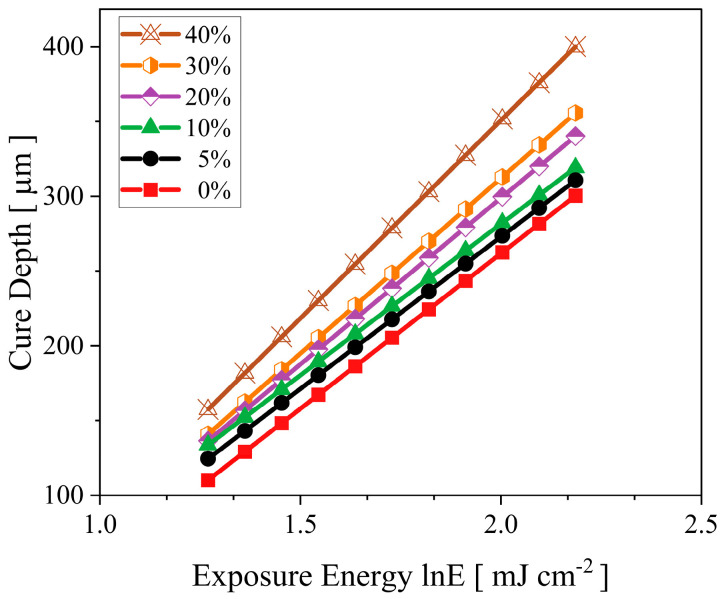
Effect of UA on the cure depth of binder mixtures at various exposure energies.

**Figure 2 polymers-15-03141-f002:**
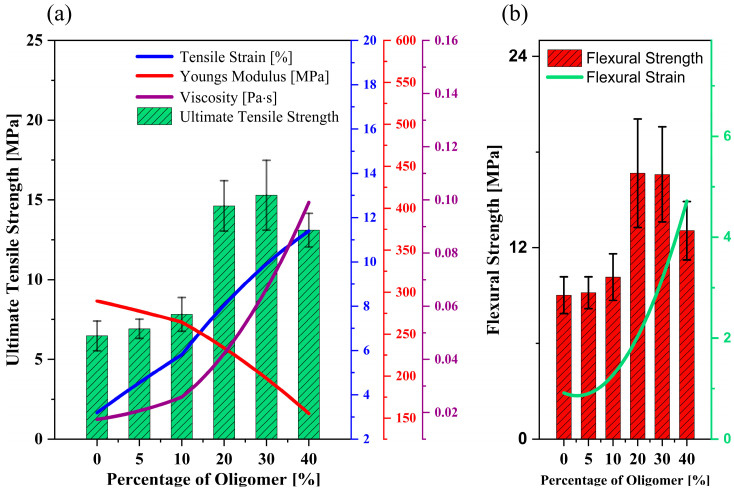
Impact of UA (Oligomer) on the (**a**) mechanical properties and viscosity of unloaded binder mixtures, (**b**) flexural strength of 75 wt.% silica loaded ceramic parts.

**Figure 3 polymers-15-03141-f003:**
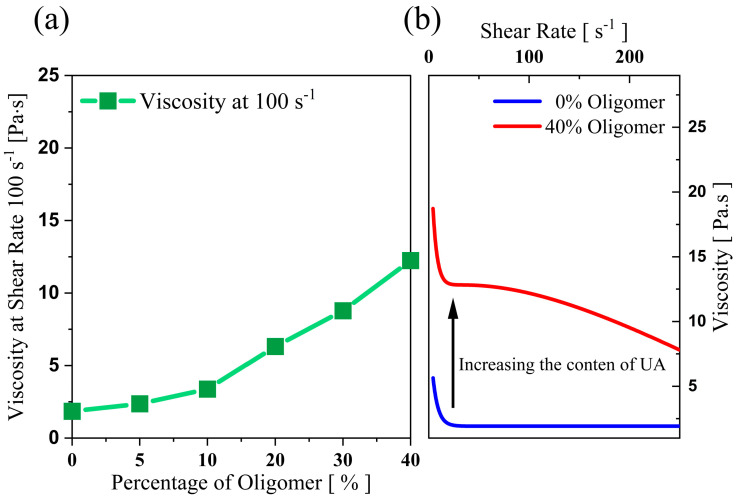
Impact of UA on the viscosity of 75 wt.% silica loaded organic medium, (**a**) viscosity of the ceramic suspension at 100 $s^{-1}$ for varying UA content (**b**) viscosity of mixtures with 0 and 40 wt.% of UA at varying shear rate, showing how UA increases the viscosity of mixtures.

**Figure 4 polymers-15-03141-f004:**
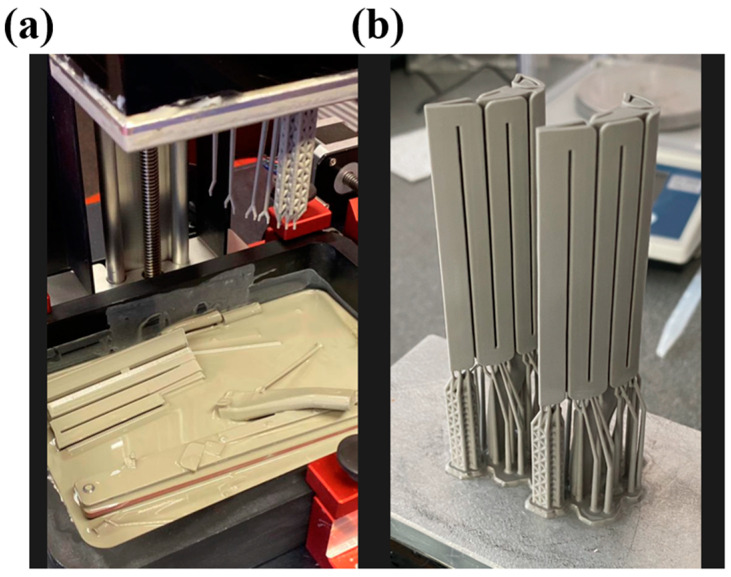
Vertically 3D-printed ceramic core model with the (**a**) B-37 and (**b**) 30 wt.% of UA-formulated mixtures, showing how UA provides substantial adhesion to the build platform of the printer.

**Figure 5 polymers-15-03141-f005:**
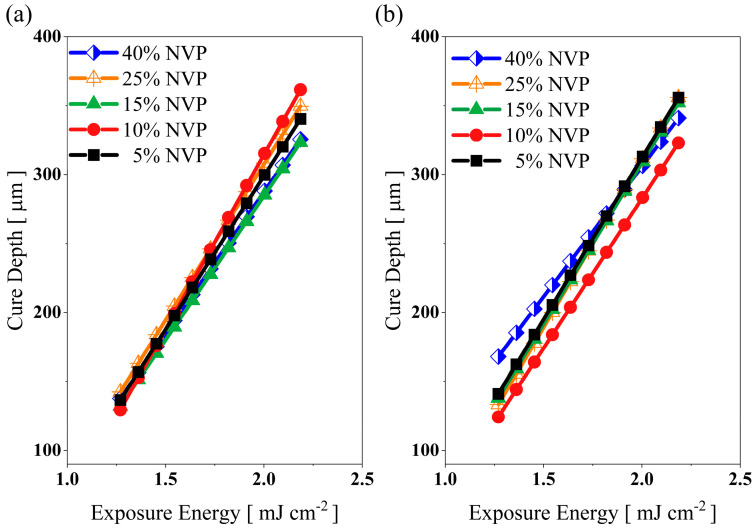
Effect of NVP on the cure depth of binder mixtures: (**a**) 20 wt.% and (**b**) 30 wt.% of UA containing mixtures at various exposure energy.

**Figure 6 polymers-15-03141-f006:**
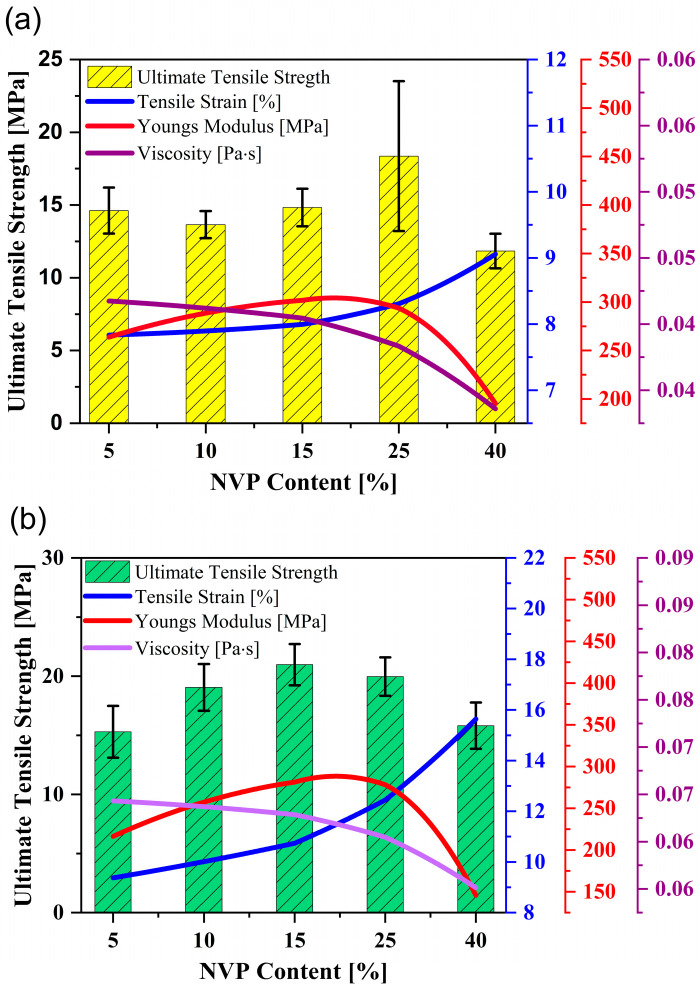
Effect of increasing the content of NVP on the mechanical properties and viscosity of (**a**) 20 wt.% and (**b**) 30 wt.% of UA formulated mixtures.

**Figure 7 polymers-15-03141-f007:**
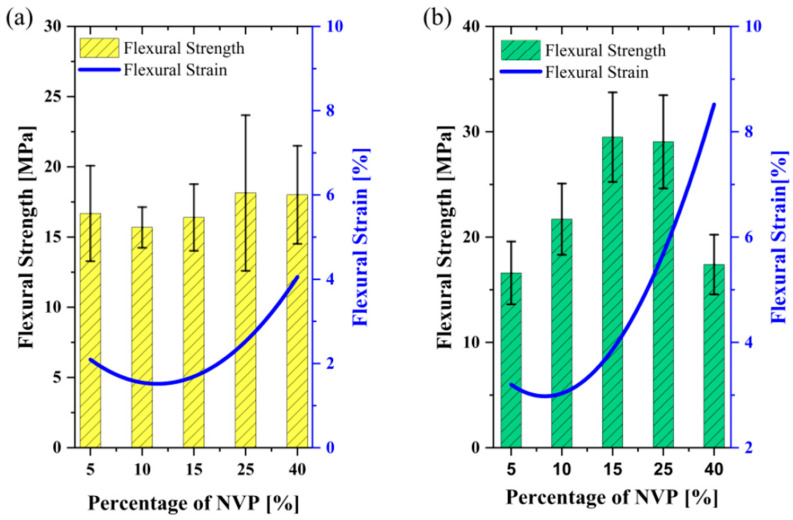
Effect of increasing the content of NVP on the mechanical properties and viscosity of (**a**) 20 wt.% and (**b**) 30 wt.% of UA formulated mixtures.

**Figure 8 polymers-15-03141-f008:**
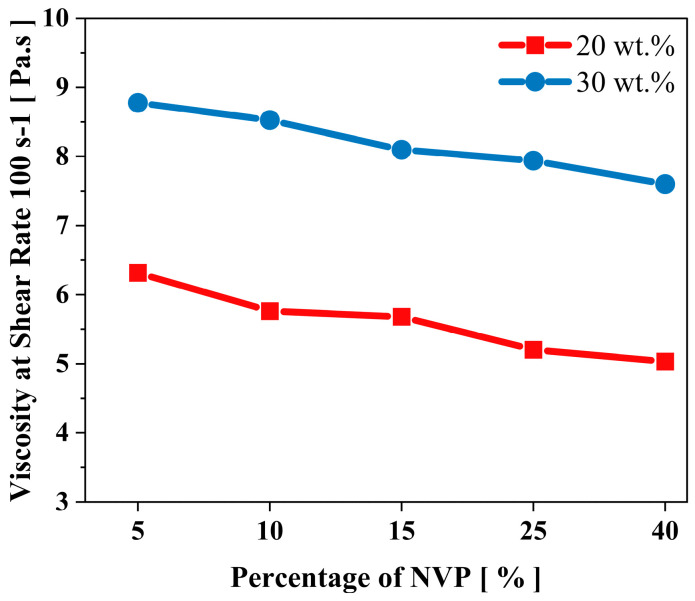
The impact of the NVP content on the viscosity of 20 and 30 wt.% of UA binder formulation at 75 wt.% ceramic loading.

**Figure 9 polymers-15-03141-f009:**
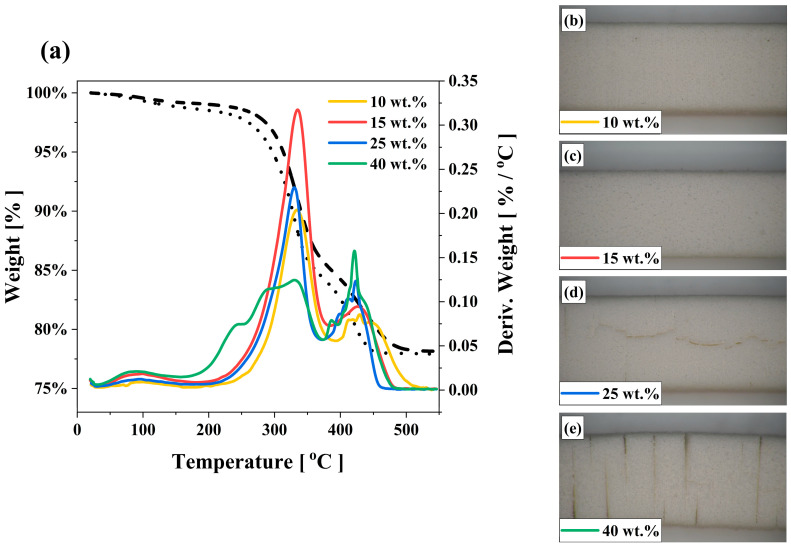
(**a**) Impact of NVP on the exothermic peaks of binder mixtures with derivative of weight curves, and microscopic images of ceramic parts after sintering showing the effect of NVP after sintering: (**b**) 10 wt.%, (**c**) 15 wt.%, (**d**) 25 wt.% and (**e**) 40 wt.% of NVP.

**Figure 10 polymers-15-03141-f010:**
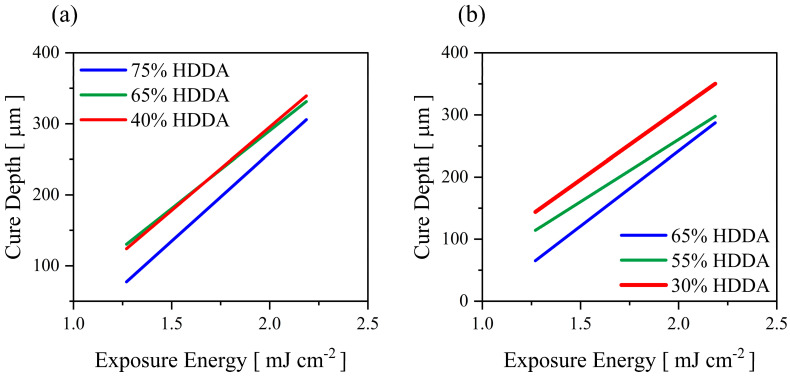
Cure depth versus exposure energy plot for (**a**) 20wt.% and (**b**) 30wt.% UA added binders under varying HDDA content.

**Figure 11 polymers-15-03141-f011:**
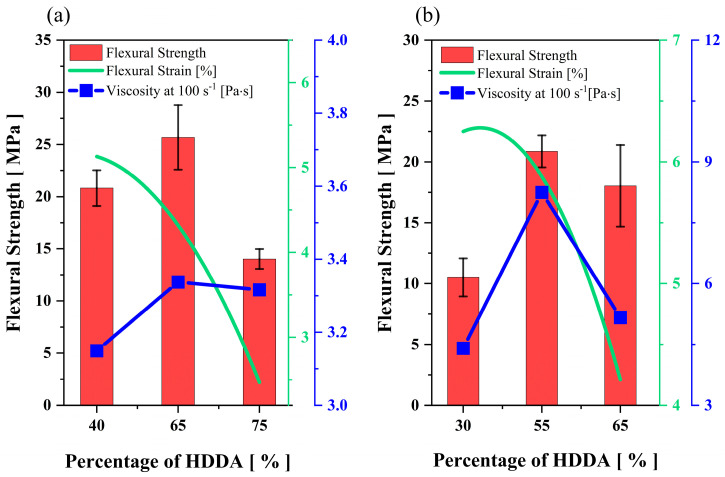
The impact of increasing the content of HDDA on the mechanical properties of ceramic parts and viscosity of ceramic suspensions for the binder systems with (**a**) 20 wt.% and (**b**) 30 wt.% of UA.

**Figure 12 polymers-15-03141-f012:**
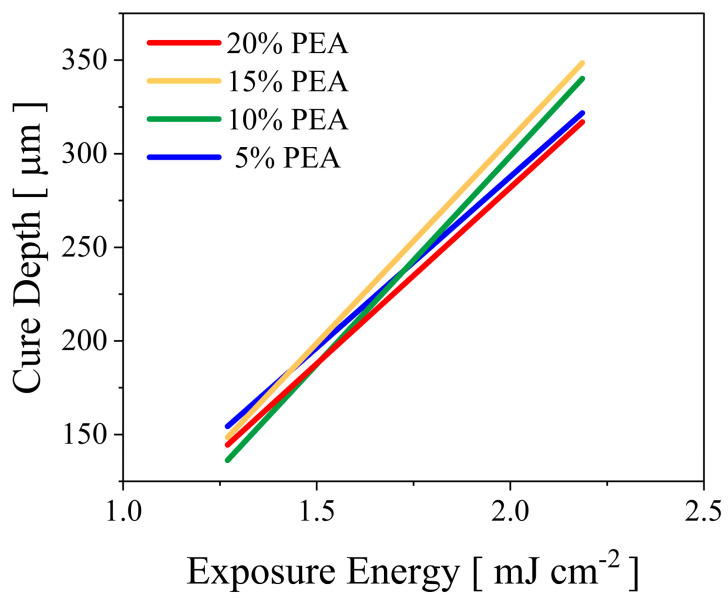
The impact of PEA on cure depth at higher exposure energy.

**Figure 13 polymers-15-03141-f013:**
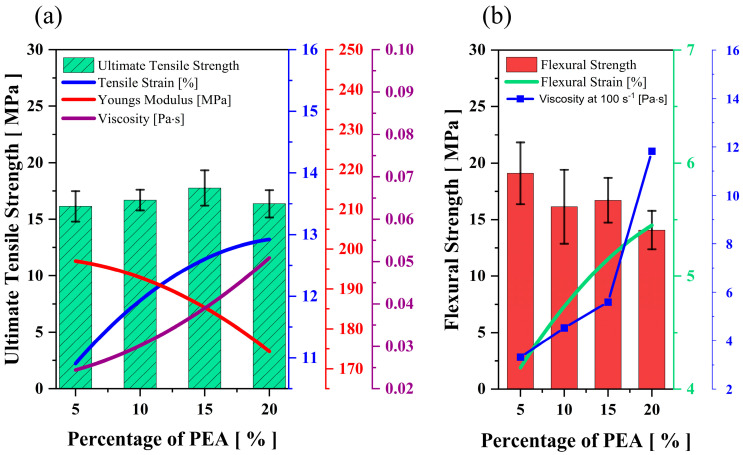
The effect of PEA on the mechanical and viscosity behaviour of (**a**) unloaded and (**b**) ceramic-loaded mixtures.

**Figure 14 polymers-15-03141-f014:**
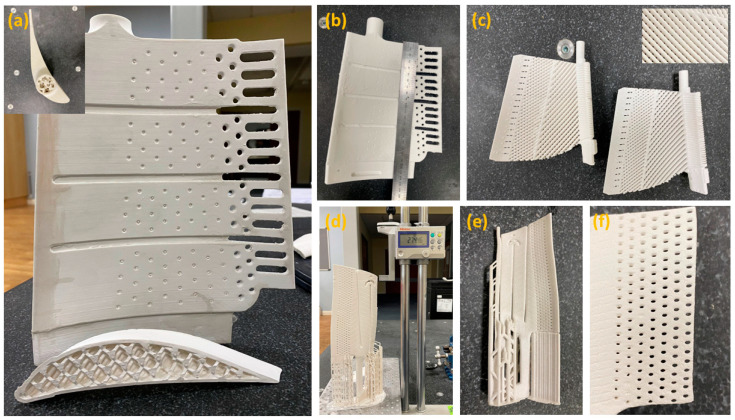
AM-manufactured ceramic core examples that cannot be achieved with conventional methods: (**a**) hollowed honeycomb structured ceramic core feature for IGT blades to improve the leaching speed after casting and (**b**) after successful sintering, (**c**) a mesh-grid structured core feature for higher cooling efficiency, (**d**) a complex vertical printed core (over 27 cm long) used in the casting of IGT blades after printing and (**e**) after sintering and (**f**) multiple small cooling holes with a less than 0.2 m of diameter after sintering.

**Table 1 polymers-15-03141-t001:** The material properties of monomers and oligomers at 25 °C.

Monomer	Label Name	Functionality	Viscosity [mPa·s]	Density [g/cm^3^]	Molecular Weight [g/mol]	Refractive Index [n20/D]
1,6 hexanediol diacrylate	HDDA	2	5–10	1.01	226	1.45
Photocentric 34	PC34	3	70–150	1.0–1.2	N/A	N/A
N-Vinyl-2-Pyrrolidone	NVP	1	2–4	1.04	111	1.48
Aliphatic urethane acrylate	UA	2	21,000	1.16	-	1.49
Polyether acrylate	PEA	3	20,000	-	-	1.50

**Table 2 polymers-15-03141-t002:** The impact of varying the content of HDDA and NVP on the mechanical and viscosity properties of the binder mixtures in 20 and 30 wt.% of UA formulated binder mixtures.

Mix No	Content of Monomer			Ultimate Tensile Strength	Strain	Young’s Modulus	Viscosity
	HDDA	UA	NVP				
	[wt.%]			[MPa]	[%]	[MPa]	[Pa·s]
1	75	20	5	14.02 ± 1.14	7.940	204.97	0.019
2	65	20	15	25.68 ± 1.28	10.28	226.32	0.017
3	40	20	40	20.82 ± 0.96	10.77	164.35	0.017
4	65	30	5	18.03 ± 1.01	10.22	179.76	0.036
5	55	30	15	20.86 ± 1.03	12.41	194.94	0.031
6	30	30	40	10.51 ± 0.71	11.96	94.91	0.031

## Data Availability

Not applicable.

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
