# Peer review of "A Systematic Study on Impact of Binder Formulation on Green Body Strength of Vat-Photopolymerisation 3D Printed Silica Ceramics Used in Investment Casting"

_polymers, 2023, doi:10.3390/polym15143141_

Round 1
Reviewer 1 Report
The authors have investigated the formulation of Vat-Photopolymerisation. The results are comprehensively presented and will contribute to the advancement of additive manufacturing. The manuscript is generally well written except for the discussion. The improvement regarding this issue among others are required before the manuscript publication.
1) The reference citations in the discussion section are lacking. The submitted version is mostly limited within itself. The manuscript will be improved if the findings are compared to or explained in the context of the published articles.
2) Lines 190-192 are wrongly copied.
3) The authors must correct Lines 515 and 639.
4) The authors should proof-read the manuscript. There are inconsistencies, particularly in the reference list. For example, the publishing company and month of publication should be left out. Abbreviation must be fully spelled out at the first time of its use.
Moderate editing of English language required
Author Response
Re: Manuscript ID: polymers-2421376  
To: MDPI Polymers Editorial Office
Dear Ms. Qu,
On behalf of all authors, I’d like to sincerely appreciate the chance to address the reviewers’ comments. We find all the comments fair and immensely helpful to make this manuscript significantly better, both in presentation and its technical/academic content. I must admit, we change it a lot and it took a lot of effort, but we all believe it made it a way paper, as one may except from Polymer Journal. We hope the revised manuscript meets the expectations of the reviewers and the editors. Below, please find our replies to address the reviewers’ comments.
Yours sincerely,
Ehsan Sabet
(PhD, MSc, BSc, MCMI, FHEA)
Additive Manufacturing & Engineering Management|Loughborough University, UK
Director & Founder| Additive Manufacturing Centre of Excellence, Derby, UK
Wolfson School of Mechanical and Manufacturing Engineering,
Loughborough University
Office Number: +44 - (0)1509 227192 ; Mobile: +44 – (0)7947279341 ;
Address: Room : TW228, Wolfson Building, Loughborough University, LE11 3TU, UK
Research Group: https://www.lboro.ac.uk/departments/meme/research/research-projects/investmentcasting/
LinkedIn: https://uk.linkedin.com/in/ehsansabet
Reviewer 1: This article could be interesting, but the presentation is difficult to follow and understand. In my opinion, the article needs extensive revision, in fact needs to be rewritten in a more accessible style. Ambiguity should be avoided, the information should be clearly presented.
Response: Thank you very much for your valuable feedback, the article has been rewritten and extensive revision has been done considering your point by point comments and suggestions. Please find below our response for each response.
- Please give full term for abbreviations when first used, followed by the abbreviation in brackets. Check the entire text, including abstract.
- All abbreviations have been checked and corrected in new manuscript.
- Line 36-41. The authors aim to describe vat photopolymerization. Vat photopolymerization comprises 2 methods: stereolithography (SLA) and digital light processing (DLP). As it seems the authors describe only DLP, but referring it to as vat photopolymerization: "light from an LCD screen [1]". Only DLP uses a screen, SLA uses a beam. Reference 1, cited by the authors is about DLP. Please clarify this notion, it's confusing.
- This has been clarified in revised manuscript.
- [3], [4] should be replaced with [3,4]. Check the entire text.
- It has been replaced and entire text checked.
- Line 73-81. In my opinion, the information regarding "commercial UV-curable resins used in polymer printing applications" is not relevant to the subject and should be removed.
- We appreciate this comment, the sentence removed and clarified in new version.
- Line 84. "a review of the existing literature". Which review, there is no reference to this statement.
- Reference to this statement has been added.
- Line 90. "previous studies [11]". It's only one study, please use the singular, not the plural. Same in line 112.
- This has been amended.
- The aim of the study should be the last sentence of the introduction.
- The aim of the study moved to the last sentence of the introduction.
- Line 125. Reference needed.
- The missing reference added in the revised version
- Table 1. The title should be corrected, the topic is not proper.
- The title has been corrected in Table 1.
- Line 148. Anycubic Photon S LCD- based 3D printer is a LCD-based SLA 3D Printer, so the DLP related information given in the Introduction should be replaced with SLA ones. Please give manufacturer's location for the printer. Same for the rheometer in line 159.
- We appreciate this comment, but this printer is a mask- or LCD-based 3D printer, and is similar to DLP printers, with a different light source. In other words, LCD and DLP printers share the same principles, material preparations, print mechanisms, and challenges. However, this has been clarified in the revised manuscript in introduction section. Clarifications have been made for the rheometer too.
- Are you 100% sure that this 3D printer is suitable for ceramics? The manufacturer's indication is for polymeric materials only.
- We appreciate these printers have been designed for printing polymer materials, and for the hobbyists. However, this study aims to show that our formulation can print ceramics on such low cost, commercial, and widely available printers, opening a significant opportunity to utilising cost-efficient printers for ceramic manufacturing. Our previous research, as well as the most recent study conducted by Leite de Camargo et al. demonstrated high potential of printing highly dense oxide ceramic parts successfully with such printers.
- Line 190-193 should be deleted, Line 232, 238. Reference needed.
- These have been fixed in revised version.
- At some point brief information on your previous published study (reference 11) should be given. It's quite confusing for the reader.
- A brief information on our previous published study has been provided in section 3.1.
- Line 241. "first two studies, the third part". Now I'm lost, which first two studies, what third part? Please be more specific and clear. Line 243. "final study" ??? Same comment as above.
- Thank you for this comment, this has been specified and clarified in section 3.1.
We hope that is clear in the revised version.
- There is no discussion of the literature. Your findings should be corroborated with other similar ones. The discussion part should be separated from the results one.
- The required discussion of literature for each section has been added, corroborated with similar studies, and they have been separated from the results in each section.
- Figure 3. There is only figure 3a, no figure 3b, it's only mentioned in the caption. Same for Figure 4b. But 3b and 4b are mentioned in the text. Please correct., Figure 10 d,e is mentioned before Figure 10 b, c this is not right, Figure 10, caption, no explanation for d. Please add, Figure 8 should be placed before figure 9. All the figures placement should be checked, their location should be in the proximity of the text, when first mentioned. Same for figure 13, 14.
- All the issues regarding Figures have been fixed in the revised document.
- Please use °C, not other versions. Check the entire text.
- The entire text has been checked and wrong versions has been corrected.
- Line 477. slurry, suspension should be explained.
- The slurry suspension has been explained in the revised document (section 2.1. and 2.2.)
- Line 515. (as seen in Error! Reference source not found.) Please delete.
- It has been fixed.
- The results part has to be reorganized, it's too long, unnecessary details should be removed and the information should be better systematized.
- This has been reorganized and some unnecessary paragraphs have been either removed or moved to supplementary document, we hope the revised version meets with exception of reviewer.
- Conclusion should be rewritten. The aim is not to be mentioned in the conclusion. No general remarks are needed, only the specific conclusions of your study. the conclusion part is too long, only a paragraph or bulleted ideas are needed.
- The conclusion section has been rewritten, as instructed.
- Line 539. "Vat-photopolymerisation is an additive manufacturing process that depends on photopolymerisation". Please rephrase. In fact, the entire first sentence should be rephrased, there's no mention that the binder is used for ceramic materials.
- This has been removed in revised section as reviewer suggested (conclusion section)
- The references list should follow the MDPI style.
- The reference list has been fixed.
- Comments on the Quality of English Language The manuscript should be further checked for language and editing errors.
- It has been checked and proofread.

Reviewer 2 Report
This article could be interesting, but the presentation is difficult to follow and understand. In my opinion, the article needs extensive revision, in fact needs to be rewritten in a more accessible style. Ambiguity should be avoided, the information should be clearly presented.
Point by point comments and suggestions:
Please give full term for abbreviations when first used, followed by the abbreviation in brackets. Check the entire text, including abstract.
Line 36-41. The authors aim to describe vat photopolymerization. Vat photopolymerization comprises 2 methods: stereolithography (SLA) and digital light processing (DLP). As it seems the authors describe only DLP, but referring it to as vat photopolymerization: "light from an LCD screen [1]". Only DLP uses a screen, SLA uses a beam. Reference 1, cited by the authors is about DLP. Please clarify this notion, it's confusing.
Line 63. [3], [4] should be replaced with [3,4]. Check the entire text.
Line 73-81. In my opinion, the information regarding "commercial UV-curable resins used in polymer printing applications" is not relevant to the subject and should be removed.
Line 84. "a review of the existing literature". Which review, there is no reference to this statement.
Line 90. "previous studies [11]". It's only one study, please use the singular, not the plural. Same in line 112.
The aim of the study should be the last sentence of the introduction.
Line 125. Reference needed.
Table 1. The title should be corrected, the topic is not proper.
Line 148. Anycubic Photon S LCD- based 3D printer is a LCD-based SLA 3D Printer, so the DLP related information given in the Introduction should be replaced with SLA ones. Please give manufacturer's location for the printer. Same for the rheometer in line 159.
Are you 100% sure that this 3D printer is suitable for ceramics? The manufacturer's indication is for polymeric materials only.
Line 190-193 should be deleted.
Line 232, 238. Reference needed.
At some point brief information on your previous published study (reference 11) should be given. It's quite confusing for the reader.
Line 241. "first two studies, the third part". Now I'm lost, which first two studies, what third part? Please be more specific and clear.
Line 243. "final study" ??? Same comment as above.
Please separate the results part from the discussion part.
Figure 3. There is only figure 3a, no figure 3b, it's only mentioned in the caption. Same for Figure 4b. But 3b and 4b are mentioned in the text. Please correct.
Figure 8 should be placed before figure 9. All the figures placement should be checked, their location should be in the proximity of the text, when first mentioned. Same for figure 13, 14.
Please use °C, not other versions. Check the entire text.
Figure 10 d,e is mentioned before Figure 10 b, c this is not right.
Figure 10, caption, no explanation for d. Please add.
Line 477. slurry, suspension should be explained.
Line 515. (as seen in Error! Reference source not found.) Please delete.
There is no discussion of the literature. Your findings should be corroborated with other similar ones. The discussion part should be separated from the results one.
The results part has to be reorganized, it's too long, unnecessary details should be removed and the information should be better systematized.
Conclusion should be rewritten.
Line 539. "Vat-photopolymerisation is an additive manufacturing process that depends on photopolymerisation". Please rephrase. In fact, the entire first sentence should be rephrased, there's no mention that the binder is used for ceramic materials.
The aim is not to be mentioned in the conclusion. No general remarks are needed, only the specific conclusions of your study. the conclusion part is too long, only a paragraph or bulleted ideas are needed.
The references list should follow the MDPI style.
The manuscript should be further checked for language and editing errors.
Author Response
Re: Manuscript ID: polymers-2421376  
To: MDPI Polymers Editorial Office
Dear Ms. Qu,
On behalf of all authors, I’d like to sincerely appreciate the chance to address the reviewers’ comments. We find all the comments fair and immensely helpful to make this manuscript significantly better, both in presentation and its technical/academic content. I must admit, we change it a lot and it took a lot of effort, but we all believe it made it a way paper, as one may except from Polymer Journal. We hope the revised manuscript meets the expectations of the reviewers and the editors. Below, please find our replies to address the reviewers’ comments.
Yours sincerely,
Ehsan Sabet
(PhD, MSc, BSc, MCMI, FHEA)
Additive Manufacturing & Engineering Management|Loughborough University, UK
Director & Founder| Additive Manufacturing Centre of Excellence, Derby, UK
Wolfson School of Mechanical and Manufacturing Engineering,
Loughborough University
Office Number: +44 - (0)1509 227192 ; Mobile: +44 – (0)7947279341 ;
Address: Room : TW228, Wolfson Building, Loughborough University, LE11 3TU, UK
Research Group: https://www.lboro.ac.uk/departments/meme/research/research-projects/investmentcasting/
LinkedIn: https://uk.linkedin.com/in/ehsansabet
Reviewer 2: The authors have investigated the formulation of Vat-Photopolymerisation. The results are comprehensively presented and will contribute to the advancement of additive manufacturing. The manuscript is generally well written except for the discussion. The improvement regarding this issue among others are required before the manuscript publication.
- The reference citations in the discussion section are lacking. The submitted version is mostly limited within itself. The manuscript will be improved if the findings are compared to or explained in the context of the published articles.
- Lines 190-192 are wrongly copied.
- The authors must correct Lines 515 and 639.
- The authors should proof-read the manuscript. There are inconsistencies, particularly in the reference list. For example, the publishing company and month of publication should be left out. Abbreviation must be fully spelled out at the first time of its use.
- Comments on the Quality of English Language Moderate editing of English language required
Response: Thank your valuable review and feedback, the discussion has been improved and rewritten, and we corroborated our finding with similar studies. All other comments have been carefully reviewed and amended in the revised version.

Round 2
Reviewer 2 Report
The manuscript is indeed improved.
Please remove reference 1 from the abstract and add it in the introduction.
Please check the Figures, there's no Figure 2. Figure 3 a and b are mentioned in the text but there's only Figure 3 a.
Figure 6 should be moved before Figure 7, as it is it overlaps Figure 7.
Author Response
Dear Review,
Thanks for your kind review and comments.
We have removed the reference from the abstract and put it in the introduction, and made the revisions to the figure as instructed.
Once again thank you for all your support and reviews.
Best wishes,
Ehsan Sabet